# Learning Deep Representations of Medical Images using Siamese CNNs with Application to Content-Based Image Retrieval

**Yu-An Chung**[*]  **Wei-Hung Weng**[†]
Computer Science and Artificial Intelligence Laboratory
Massachusetts Institute of Technology, Cambridge, MA 02139
{andyyuan,ckbjimmy}@mit.edu

## Abstract

Deep neural networks have been investigated in learning latent representations of medical images, yet most of the studies limit their approach in a single supervised convolutional neural network (CNN), which usually rely heavily on a large scale annotated dataset for training. To learn image representations with less supervision involved, we propose a deep Siamese CNN (SCNN) architecture that can be trained with only binary image pair information. We evaluated the learned image representations on a task of content-based medical image retrieval using a publicly available multiclass diabetic retinopathy fundus image dataset. The experimental results show that our proposed deep SCNN is comparable to the state-of-the-art single supervised CNN, and requires much less supervision for training.

## 1 Introduction

Effective feature extraction and data representation are key factors to successful medical imaging tasks. Researchers usually adopt medical domain knowledge and ask for annotations from clinical experts. For example, using traditional image processing techniques such as filters or edge detection techniques to extract clinically relevant spatial features from images obtained by different image modalities, such as mammography [Tsochatzidis et al., 2017], lung computed tomography (CT) [Dhara et al., 2017], and brain magnetic resonance imaging (MRI) [Jenitta and Ravindran, 2017]. The handcrafted features with supervised learning using expert-annotated labels work appropriately for specific scenarios. However, using predefined expert-derived features for data representation limits the chance to discover novel features. It is also very expensive to have clinicians and experts to label the data manually, and such labor-intensive annotation task limits the scalability of learning generalizable medical imaging representations.

To learn efficient data representations of medical images, researchers recently have used different deep learning approaches and applied to various medical image machine learning tasks, such as image classification [Esteva et al., 2017, Gulshan et al., 2016], image segmentation [Havaei et al., 2017, Guo et al., 2017], or content-based image retrieval (CBMIR) [Litjens et al., 2017, Sun et al., 2017, Anavi et al., 2016, Liu et al., 2016, Shah et al., 2016].

CBMIR is a task that helps clinicians make decisions by retrieving similar cases and images from the electronic medical image database [Müller et al., 2004] (Figure 1). CBMIR requires expressive data representations for knowledge discovery and similar image identification in massive medical image databases, and has been explored by different algorithmic approaches [Kumar et al., 2013, Müller et al., 2004].

---

[*]Co-first author

[†]Co-first author, corresponding author

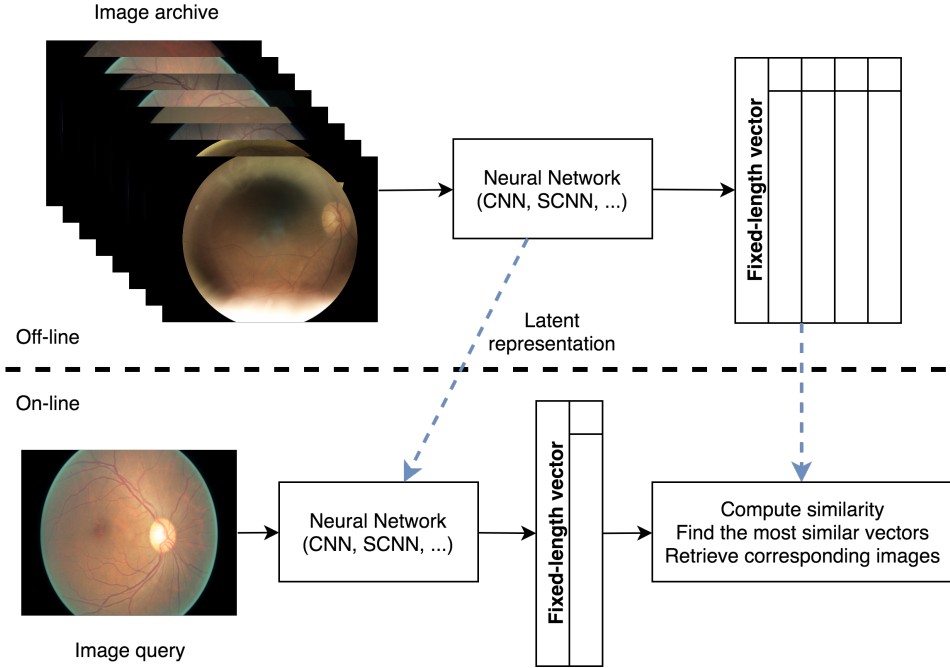

Figure 1: Overview of content-based medical image retrieval.

However, the previous works of CBMIR focused more on using shallow learning algorithms, or combining single pre-trained CNN structure with other techniques [Litjens et al., 2017, Sun et al., 2017, Anavi et al., 2016, Liu et al., 2016, Shah et al., 2016], which relies heavily on manually annotated, high-quality ground truth labeling.

To mitigate these issues, we proposed a deep Siamese CNN (SCNN) that can learn fixed-length latent image representation from solely image pair information in order to reduce the dependency of using actual class labels annotated by human experts [Bromley et al., 1994]. We then evaluated the learned image representations on the task of CBMIR using a publicly available diabetic retinopathy (DR) fundus image dataset. We compared the image representations learned by the proposed deep SCNN with the single pre-trained supervised CNN architecture [He et al., 2016].

The architecture of the proposed deep SCNN is illustrated in Figure 2. The deep SCNN learns to differentiate an image pair by evaluating the similarity and relationship between the given images. Each image in the image pair is fed into one of the identical CNN, and the contrastive loss is computed between two outputs of CNNs. The model is an end-to-end structure to obtain a latent representation of the image, which can be used for further CBMIR task.

The main contributions of this work are that we propose an end-to-end deep SCNN model for learning latent representations of medical images with minimal expert labeling efforts by reducing the multiclass problem to binary class learning problem, and apply them in the task of CBMIR using retina fundus images as a proof of concept. Experimental results show that SCNN's performance is comparable to that of the state-of-the-art CBMIR method using single supervised pre-trained CNN, but requires much less supervision for training.

## 2   Related Work

Recently, deep neural networks have been adopted in medical image learning tasks and yield the state-of-the-art performance in many medical imaging problems [Litjens et al., 2017]. Using deep neural networks allows automatic feature extraction and general, expressive representation learning for different computer vision tasks [Bengio et al., 2013], including machine learning tasks of medical imaging. After Krizhevsky et al. [2012] yielded a breakthrough performance using deep convolutional neural network (CNN) for ImageNet challenge [Deng et al., 2009], supervised learning with CNN

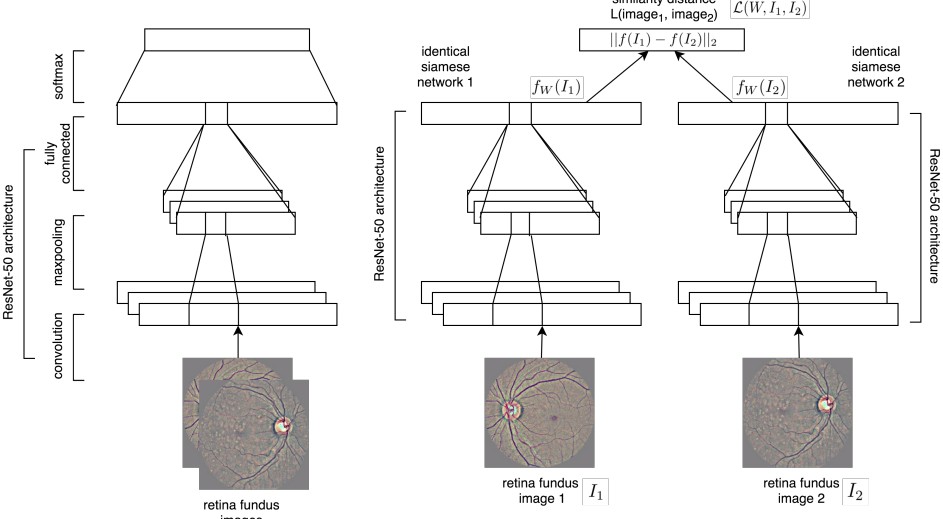

Figure 2: Architecture of used model in the study. (a) Single convolutional neural network, and (b) proposed deep Siamese convolutional neural networks.

architecture has become a general structure for visual tasks. For medical image, researchers mainly use CNN, stacked autoencoder [Cheng et al., 2016], and restricted Boltzmann machine [Brosch et al., 2013] for different tasks such as classification [Esteva et al., 2017, Gulshan et al., 2016], segmentation [Havaei et al., 2017, Guo et al., 2017], image generation and synthesis [Nie et al., 2017, Van Nguyen et al., 2015], image captioning [Moradi et al., 2016, Shin et al., 2015], and CBMIR [Sun et al., 2017, Anavi et al., 2016, Liu et al., 2016, Shah et al., 2016].

Deep learning is not yet widely adopted in CBMIR except for few studies on lung CT [Sun et al., 2017], prostate MRI [Shah et al., 2016], and X-ray image [Anavi et al., 2016, Liu et al., 2016]. Sun et al. [2017] applied CNN with residual network to retrieve lung CT images. Shah et al. [2016] adopted CNN with hashing-forest method for prostate MRI image retrieval. Anavi et al. [2016] used a five-layered pre-trained CNN, extracted the image representation in the fully-connected layer, integrated textual metadata, and fed into a support vector machine (SVM) classifier for distance measurement. Liu et al. [2016] combined three-layer CNN with Radon barcodes to retrieve images from 14,410 chest X-ray images. Different from the previous works, our proposed approach elaborates the capability of deep SCNN to reduce the labeling effort by using only binary image pair information, rather than the exact multiclass labeling.

## 3 Approach

Our method utilized end-to-end deep SCNN architecture to learn fixed-length representations from images with minimal expert labeling information [Bromley et al., 1994].

### 3.1 Deep Siamese Convolutional Neural Networks

Deep SCNN architecture is a variant of neural network that can find the relationship and similarity between the input objects. It has multiple symmetric subnetworks tying the same parameters and weights and updating mirrorly, and cojoining at the top by an energy function. Siamese neural networks were originally designed to solve signature verification problem of image matching [Bromley et al., 1994]. It has also been used for one-shot image classification [Koch et al., 2015].

We construct deep SCNN for learning fixed-length representations using two identical CNNs sharing the same weights. In our experiment, each identical CNN was built using ResNet-50 architecture with the ImageNet pre-trained weight [He et al., 2016]. We used 25% dropout for regularization to reduce overfitting and adopted batch normalization [Srivastava et al., 2014, Ioffe and Szegedy, 2015]. The rectified linear units (ReLU) nonlinearity was applied as the activation function for all layers,

and we used adaptive moment estimation (Adam) optimizer to control learning rate [Kingma and Ba, 2014]. The similarity between images was calculated by Euclidean distance, and we defined loss function by computing the contrastive loss [Hadsell et al., 2006], which can be presented in the equation:

$$\mathcal{L}(W, I_1, I_2) = \mathbf{1}(L = 0)\frac{1}{2}D^2 + \mathbf{1}(L = 1)\frac{1}{2}[\max(0, margin - D)]^2$$

, where $I_1$ and $I_2$ are a pair of retina fundus images fed into each of two identical CNNs. $\mathbf{1}(\cdot)$ is a indicator function to show that whether two images have the same label, where $L = 0$ represents the images have the same label and $L = 1$ represents the opposite. $W$ is the shared parameter vector that neural networks will learn. $f(I_1)$ and $f(I_2)$ are the latent representation vectors of input $I_1$ and $I_2$, respectively. $D$ is the Euclidean distance between $f(I_1)$ and $f(I_2)$, which is $||f(I_1) - f(I_2)||_2$.

Comparing to the single supervised CNN which uses multiclass information, the SCNN transforms the multiclass problem to binary classification learning problem.

### 3.2 Baseline

In this study, we compared end-to-end deep SCNN with an end-to-end single supervised ResNet-50 architecture. We implemented all neural networks with Keras.

### 3.3 Evaluation

We used two metrics to evaluate the performance of CBMIR, (1) mean reciprocal rank (MRR),

$$MRR = \frac{1}{Q}\Sigma_{i=1}^{Q}\frac{1}{rank_i}$$

where $Q$ is the query size and $rank_i$ means that the rank of the real first-ranked item in the $i$-th query and (2) mean average precision (MAP),

$$MAP = \frac{1}{Q}\Sigma_{i=1}^{Q}AveP$$

where $AveP$ is the area under the precision-recall curve.

## 4 Experiment

We conducted experiments and trained our model on a subset of DR fundus image dataset to demonstrate the capability of SCNN architecture. We then analyzed and evaluated the performance of learning representations and CBMIR between different approaches.

### 4.1 The Diabetic Retinopathy Fundus Image Dataset

As a severe complication of diabetic mellitus, DR is a common cause of blindness around the world, especially in the developed countries due to the high prevalence of diabetes mellitus. Screening and detection of early DR are therefore critical for disease prevention. DR fundus image database is collected, maintained and released by EyePACS, a free platform for retinopathy screening, and released as the dataset for Kaggle competition. We used the full training set of Kaggle Diabetic Retinopathy Detection challenge with 35,125 fundus images. Five clinical severity labels from normal/healthy to severe (labeled as 0, 1, 2, 3, and 4 in the dataset) were given by experts and used for the single CNN approach in the study.

### 4.2 Data Preprocessing and Augmentation

To remove variations caused by camera and lighting conditions of different fundoscopes, we rescaled and normalized all images to the same radius, subtracted the local average color and preserved the central 90% images for boundary effect, and resized the images to $224 \times 224$ pixels.

There are 25,809 images in the largest class (normal) and only 708 images in the smallest class (most severe DR) (Figure 3). The label distribution shows the severe class imbalance in our dataset.

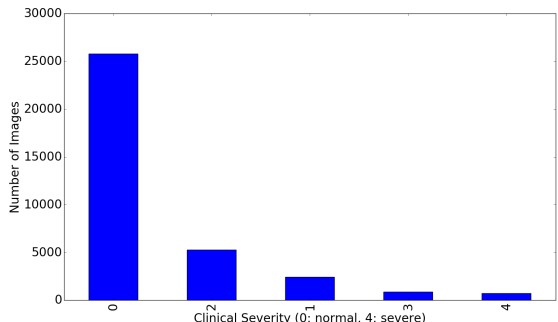

Figure 3: Expert-anotated label distribution.

To handle class imbalance, we augmented the numbers of images of all classes to the same as the largest class by randomly selected images from the minor classes and performed Krizhevsky style random offset cropping [Krizhevsky et al., 2012], random horizontal and vertical flipping, Gaussian blurring and rotation between 0° and 360°. The original and augmented images were pooled together and split into 70% train and 30% test data based on stratification of class labels.

### 4.3 Learning Latent Representations

For both single supervised CNN and deep SCNN architecture, we extracted the last bottleneck layer as our latent image representation. Principal component analysis was first adopted to reduce the feature dimension to 50, then t-Distributed Stochastic Neighbor Embedding (t-SNE) was applied to further reduce the dimension to two [Maaten and Hinton, 2008] for visualizing the latent feature embeddings.

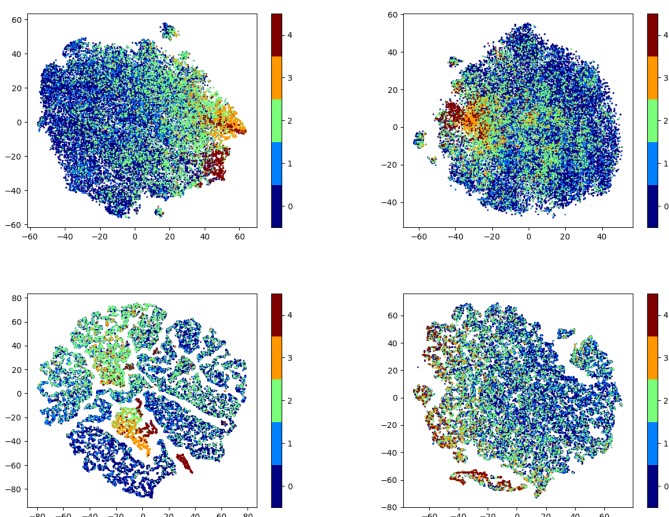

Figure 4: t-Distributed Stochastic Neighbor Embedding (t-SNE) visualizations for the distribution of learned retina fundus image representation embedding in the two-dimension vector space. (upper left) The embedding from the third-to-last layer of single CNN. (upper right) The embedding from the second-to-last layer of single CNN. (lower left) The embedding from the last softmax layer of single CNN. (lower right) The embedding from the last layer of deep SCNN. Colors represent the real expert-labeled severity of DR, where blue indicates normal/healthy cases and dark red represents severe DR cases.

In Figure 4, we demonstrate the data distribution of image representations extracted from different layers of CNN and SCNN. A clear clinically interpretable severity transition from healthy cases (label 0) to severe disease (label 3 and 4) is shown in the t-SNE visualization of baseline and proposed representations, which indicates that the learned representations are reliable.

The softmax (last) layer of CNN learned the tighter representation using multiclass information. Comparing to the softmax layer of CNN, the last layer of deep SCNN and the third-to-last and second-to-last layers of CNN learned the sliding scale representations. However, the ground truth multiclass labels given by experts are arbitrary. The real DR condition is progressive gradually instead of having a strict cutting-off boundary between each stage of severity. The sliding scale representations are therefore more desirable to express the real DR pathology.

## 4.4 Content-Based Medical Image Retrieval

In the experiment of CBMIR in DR, we compared the performance of our proposed deep SCNN model with the corresponding single supervised pre-trained ResNet-50 architecture, and performed image retrieval on few sample queries of DR fundus images (Figure 5).

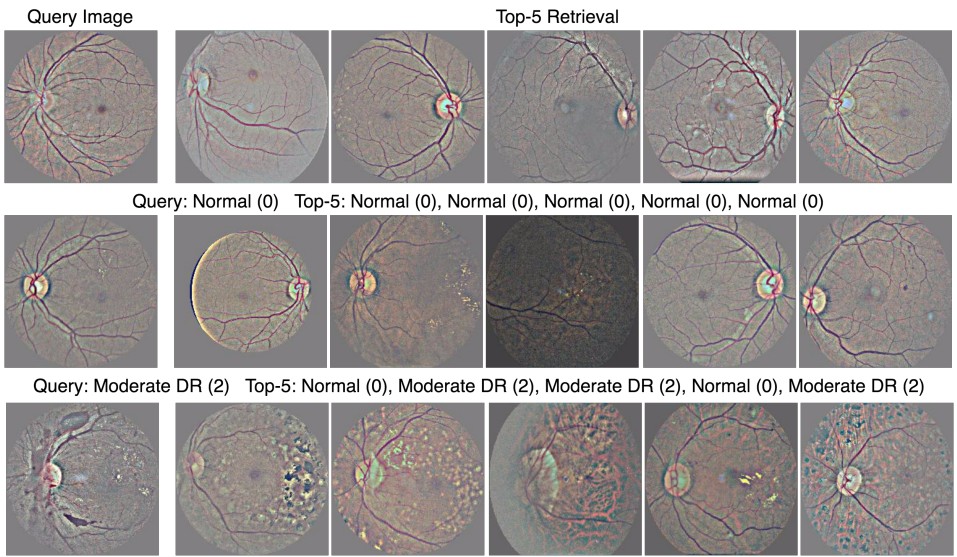

Figure 5: Three samples of image retrieval after correct image preprocessing.

Table 1 shows that the proposed deep SCNN architecture yielded the comparable performance of image retrieval using minimal expert labeling with only binary image pair information, compared to the single CNN model, which requires exact expert labeling multiclass information. Considering the preferred sliding scale representations, the representation learned by deep SCNN with binary labeling outperformed those learned by either third-to-last or second-to-last layers of single CNN.

Table 1: Performance measurement of CBMIR using latent representations from single pre-trained CNN or Siamese CNNs (SCNN)

| Layer | MAP | MRR |
|---|---|---|
| CNN (third-last) | 0.6209 | 0.7608 |
| CNN (second-last) | 0.6369 | 0.7691 |
| CNN (softmax) | 0.6673 | 0.7745 |
| SCNN (last layer) | 0.6492 | 0.7737 |

To evaluate the quality of CBMIR using deep Siamese CNNs, we made real queries and extracted similar images for clinical qualitative evaluation. In Figure 5, we show three sample queries with different DR severity and the top five corresponding retrieved examples using deep SCNNs.

The first query using normal/healthy fundus image yielded exactly the fundus images with the same expert label. There are few inconsistencies while using the image of moderate or severe DR as query inputs. However, in Figure 5 we are able to see that the inconsistencies result from either artifacts of fundus images or ambiguous diagnosis of DR severity. For example, the third retrieval of the query (third row in Figure 5) using severe DR images show that the artifact of original image leads to incorrect classification. The fourth retrieval of severe DR query is the image on the borderline between moderate and severe DR, which may also be a challenging case for some clinicians.

In general, the deep SCNNs can identify and extract fundus images which have the same expert-annotated label as the input query, or the images with very similar patterns but has deviated severity label.

## 5   Conclusions

In this paper, we have presented a new strategy to learn latent representations of medical images by learning an end-to-end deep SCNN, which only requires binary image pair information. We performed the experiment on the CBMIR task using publicly DR image dataset and demonstrates that the performance of deep SCNN is comparable to the commonly used single CNN architecture, which requires actual multiclass expert labeling that is expensive in the medical machine learning tasks. Future investigation will focus on performing experiments on different network architectures, other ranking metrics for evaluation such as recall on top-N, and applying the proposed method to different medical image datasets, such as chest X-ray imaging.

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
