# OpenReview forum: "Learning Deep Representations of Medical Images using Siamese CNNs with Application to Content-Based Image Retrieval"
_MIDL.amsterdam/2018/Conference — Submitted to MIDL 2018_

### Review · AnonReviewer2 · 2018-05-02
**Approach not suitable for this dataset (although potentially interesting for others)**

**Rating:** 2
**Confidence:** 3

**Review:**

In general, the work tries to apply weakly supervised CNN training to content-based image retrieval, which is one of the main applications of AE or GANs. Here, a Siamese approach is chosen, where two identical ResNet-50 CNNs are used to compute latent features from two images, and a contrastive loss is trained to compute a binary distance from the Euclidean distance of the latent vectors.
I have the following points of criticism for this work:
1) The authors claim that the method achieves "comparable" performance to supervised CNNs (a plain ResNet-50), but requires "much less supervision". However, they do not describe how the binary image pair information should be created in practice. (In the experiments in the paper, they are derived from multi-class labels, so the expensive labeling has already happened and could be used for supervised training, which further increases performance.)
2) Furthermore, the chosen task and dataset does not have categorical labels, but ordered ones (severity of DR). This fact is even used for arguing about the visual impression of the t-SNE plots in 4.3. This would in my opinion speak for a different loss function, where the distance between categories is not binary.
In summary, I do not think that the dataset is a good fit for the approach. CBIR is a very relevant problem for medical images, but more care has to be taken in order to define an appropriate task / setting.
Other suggestions / comments:
- I would not use preprocessed images in Fig. 5, where an example CBIR query is depicted. (And what does "the *correct* preprocessing" refer to? What would be an incorrect one?)
- A point is made that the SCNN outperforms "either third-to-last or second-to-last layers of single CNN" (referring to Table 1), but that comparison looks strange to me. Wouldn't you *expect* the CNN to work worse if you remove more layers than trained with? Shouldn't you also strip the SCNNs for fairness, then?
- "A clear clinically interpretable severity transition … is shown in the t-SNE visualization" is a too strong claim, in particular given the fact that the separation is not perfect at all (zooming in, I can see dots of all colors in all areas, only with different densities) and that the authors also claim in a different paragraph that "the ground truth multiclass labels given by experts are arbitrary".
- The data augmentation scheme does not fit the problem. Horizontal flipping sounds good, but vertical flipping and arbitrary 0...360 deg. rotations are not useful for DR images. Furthermore, since the augmentation is only used to "fill up" minor classes, their statistics will be very different from the majority class.
- The English is a little weak, which sometimes made it harder / more time-consuming to understand the text, but it was still readable. (For instance, the third sentence in the introduction, which is very long, is missing any active verb, so one would read it several times, trying to identify subject / predicate / object.)

**Special Issue:**

No

---

### Review · AnonReviewer3 · 2018-05-07
**Interesting Siamese approach for medical images (CBMIR), but limited quantitative results**

**Rating:** 4
**Confidence:** 2

**Review:**

The authors proposed a similarity-based approach to generate representations of medical images, in the context of CBMIR.
The proposed method consists in learning a representation modeled by a siamese CNN, trained via the optimization of a target similarity metric between pairs of images.
The method was evaluated on the task of CBMIR, by comparing the representations of DR fundus images learned by a baseline ResNet trained to classify images by their grade and the proposed approach.
Quantitative results show query scores aligned for the two approaches.
The authors made a tSNE analysis of the learned representations, and show that by their method produces a desirable "sliding-scale" property, which is lost in the late layers in the case of the baseline approach.

------

Strengths of the paper:
- Originality of the similarity-based approach for this application.
- Well conducted comparative analysis.
- Interesting interpretation of the resulting 2D embeddings.
- Relevant references.

Weaknesses of the paper:
- Limited quantitative results, no relative improvement from the baseline.
- Unclear decisions made for the design of the method (see remarks below).

Other remarks:
- Very important decisions regarding the hyper-parameters are missing and hampers the replicability of the method.
- What was the balance of the pairs of images at training time?
- The similarity metric used for the query process is not clearly expressed.
- Why choosing an l2-based similarity metric compared to the l1-based distance in [Koch et. al, 2015] the authors referenced?
- In figure 2, the labels (a) and (b) evoked in the caption are not used in the figure.

------

I acknowledge the work done by the authors and think it should be presented at MIDL as the siamese approach is new in this context, and it is valuable as it gives more attention to this type of unsupervised approaches for applications in medical image analysis. The paper gives directions for improvement in CBMIR and opens discussion for further applications.

**Special Issue:**

Yes

---

### Review · AnonReviewer1 · 2018-05-09
**representation learning with siamese CNNs for image retrieval**

**Rating:** 2
**Confidence:** 3

**Review:**

This is interesting work utilising a siamese architecture for learning representations in the context of image retrieval. However, this is not entirely novel and references to works using Fisher vectors could be added and discussed (e.g., see https://arxiv.org/abs/1702.00338).

There are quite a few details missing about the training procedure, data splitting, and selection of hyper-parameters.

I am not sure about the claim that the proposed method simplifies the data labeling task. The proposed binary problem still requires the full availability of multi-class annotations, no?

There is also no comparison to alternative CBIR methods, which makes it difficult to assess how well the system works (or how difficult the problem is). The comparison to a single supervised architecture is good, but not sufficient for this application, as there are many other methods for CBIR (in particular, in the non-medical domain).

It would also be good to add class-wise results, and precision and recall in the quantitative evaluation.

**Special Issue:**

No

---

### Decision · Program_Chairs · 2018-05-15
**Paper43 Acceptance Decision**

Reject